# Complete Primary Pathological Response Following Neoadjuvant Treatment and Radical Resection for Pancreatic Ductal Adenocarcinoma

**DOI:** 10.3390/cancers16020452

**Published:** 2024-01-20

**Authors:** Kai Tai Derek Yeung, Joseph Doyle, Sacheen Kumar, Katharine Aitken, Diana Tait, David Cunningham, Long R. Jiao, Ricky Harminder Bhogal

**Affiliations:** 1Royal Marsden Hospital, London SW3 6JJ, UK; k.yeung@imperial.ac.uk (K.T.D.Y.);; 2Imperial College London, London SW7 2BX, UK; 3The Institute of Cancer Research, London SW3 6JB, UK

**Keywords:** pancreatic cancer, neoadjuvant treatment, chemotherapy, chemoradiation, complete pathological response

## Abstract

**Simple Summary:**

Patients diagnosed with advanced pancreatic cancer are commonly treated with pre-operative chemotherapy and/or chemoradiotherapy. The aim of this study is to describe a unique group of patients treated at a single tertiary institution who had undergone pre-operative chemotherapy and/or chemoradiotherapy followed by surgical resection and were found to have no residual active cancer with the resected primary tumour site.

**Abstract:**

Introduction: Neoadjuvant treatment (NAT) for borderline (BD) or locally advanced (LA) primary pancreatic cancer (PDAC) is now a widely adopted approach. We present a case series of patients who have achieved a complete pathological response of the primary tumour on final histology following neoadjuvant chemotherapy +/− chemoradiation and radical surgery. Methods: Patients who underwent radical pancreatic resection following neoadjuvant treatment between March 2006 and March 2023 at a single institution were identified by retrospective case note review of a prospectively maintained database. Results: Ten patients were identified to have a complete primary pathological response (ypT0) on postoperative histology. Before treatment, five patients were considered BD and five were LA according to National Comprehensive Cancer Network guidelines. All patients underwent staging Computed Tomography (CT) and nine underwent ^18^Fluorodeoxyglucose Positron Emission Tomography (^18^FDG-PET/CT) imaging, with a mean maximum standardized uptake value (SUVmax) of the primary lesion at 6.14 ± 1.98 units. All patients received neoadjuvant chemotherapy, and eight received further chemoradiotherapy prior to resection. Mean pre- and post-neoadjuvant treatment serum Ca19-9 was 148.0 ± 146.3 IU/L and 18.0 ± 18.7 IU/L, respectively (*p* = 0.01). The mean duration of NAT was 5.6 ± 1.7 months. The mean time from completion of NAT to surgery was 13.1 ± 8.3 weeks. The mean lymph node yield was 21.1 ± 10.4 nodes, with one patient found to have 1 lymph node involved. All resections were reported to be R0. The mean length of stay was 11.8 ± 6.2 days. At the time of analysis, one death was reported at 35 months postoperatively. Two cases of recurrence were reported at 16 months (surgical bed) and 33 months (pulmonary). All other patients remain alive and under active surveillance. The current overall survival is 26.6 ± 20.7 months and counting. Conclusions: Complete primary pathological response is uncommon but possible following neoadjuvant treatment in patients with PDAC. Further work to identify the common denominator within this unique cohort may lead to advances in the therapeutic approach and offer hope for patients diagnosed with borderline or locally advanced pancreatic ductal adenocarcinoma.

## 1. Introduction

Pancreatic ductal adenocarcinoma (PDAC) is responsible for 5.2% of all cancer deaths in the United Kingdom [1]. Over 8000 cases are diagnosed yearly, with a five-year survival rate that has been unchanged over the last few decades at 5%. The presentation of pancreatic cancer is often late, with clinically apparent symptoms present only at the latter stages of disease, at which point treatment strategies including the potential for curative resection are limited. Locally advanced (LA) or borderline resectable (BD) PDAC is classified as per the National Comprehensive Cancer Network (NCCN) guidelines [2]. The classification is based upon the tumour’s association with adjacent venous, arterial, and retroperitoneal structures as well as the potential to resect and reconstruct the associated vascular structures. 

With resectable disease, surgical resection followed by adjuvant systemic anti-cancer therapy remains the mainstay of treatment [3]. However, even with a complete resection (R0), which is the only chance for a cure, the overall survival (OS) is only around 3 years [4]. 

The use of systemic anti-cancer and chemoradiation treatment as neoadjuvant treatment (NAT) has gained momentum recently and provides a means of downstaging BD or LA disease, allowing for a potential route to surgical resection. The perceived benefits of NAT include a reduction in involved margins, the treatment of micro-metastatic disease, and can also be a test of tumour biology prior to the consideration of surgical resection. There are, however, concerns that aggressive NAT may led to the deconditioning of patients and, subsequently, a reduced resection rate [5]. So, while the role of neoadjuvant systemic treatment in the resectable setting is yet to be proven in the clinical trial setting [6], the ESPAC-5 trial results supported the use of neoadjuvant chemotherapy in BD PDAC [7], conferring a survival benefit, which is now incorporated into clinical practice worldwide. 

Subsequent recent clinical trials, including the Alliance A021501 trial comparing mFOLFIRINOX with hypofractionated RT to mFOLFIRINOX alone [8] and the PREOPANC-2 trial comparing neoadjuvant FOLFIRINOX with neoadjuvant gemcitabine- based CRT [9] prior to surgery, reported no difference in OS with their respective comparison groups as treatment for PDAC. However, the long-term results of PREOPANC-1 trial did eventually demonstrate neoadjuvant chemoradiotherarpy followed by surgery and adjuvant gemcitabine conferred improved long-term OS compared to upfront surgery alone [10].

Nevertheless, the resection of LA PDAC following downstaging with NAT in appropriately selected patients has been shown to lead to a survival of almost 3 years [11,12]. The criteria for proceeding to surgical resection following NAT differ amongst institutions but in general require a sustained radiological and metabolic response following NAT. In fact, following NAT regimens, some tumours have even been reported to demonstrate a complete response (ypT0) upon histopathological evaluation of resected specimens. Pathological complete response (pCR) is reported to be significantly associated with improved oncological outcomes in other cancers [13,14] but remains rare with PDAC. The first case of pCR following neoadjuvant treatment and surgical resection in LA PDAC was reported in 2007 [15]. Since then, there have been reports within the literature that patients with BD/LA PDAC treated with NAT followed by surgical resection and found to have pCR can have an overall survival in the range of 43 to 76 months [16,17,18]. 

We hereby present a single institution case series of patients who have achieved a complete pathological response of the primary PDAC tumour following neoadjuvant chemotherapy and/or chemoradiation and radical surgery.

## 2. Methods

This single-institution case series included patients who have achieved complete primary pathological response—ypT0 following neoadjuvant chemotherapy and/or chemoradiation and radical surgery for PDAC between July 2006 and March 2023 at our institution.

All patient treatment plans were discussed and endorsed at each stage of treatment by a dedicated pancreatic multidisciplinary team (MDT). Patients also only underwent radical resection after fitness for major resection was reviewed by a consultant-led anaesthetic pre-assessment including cardiopulmonary exercise testing (CPEX). Patients for this specific series were identified by a retrospective electronic case note review of a prospectively maintained database.

The primary endpoint of this study was overall survival (OS) from day of surgery, and secondary endpoints were to describe patient outcomes after surgical resection. Surgical complications were graded using the Clavien–Dindo classification [19]. This study was approved by the Institutional Review Board of the Royal Marsden NHS Foundation Trust, London, United Kingdom. All statistical analysis and graphical representations were conducted and produced with R (v 4.3.2) run in R Studio (v2023.09.1 + 494). All median values are listed with respective ranges and mean values with respective standard deviation.

## 3. Results

### 3.1. Patient Demographics

Ten patients with a median age of 64 years (range: 50 to 71 years) were included in this case series (Table 1). The mean BMI was 22.89 ± 3.48 kg/m^2^. All patients underwent triple-phase computerised tomography (CT) scanning, and all but one patient underwent ^18^FDG-PET/CT as part of routine pre-operative diagnostic and staging investigations at the Royal Marsden Hospital. Staging and assessment as per the NCCN guidelines of PDAC were performed by a specialist pancreatic radiologist at dedicated pancreatic MDT. PDAC was histologically proven by endoscopic or endoscopic ultrasound-guided biopsy in all cases prior to the commencement of systemic anti-cancer treatment. Five patients (50%) had BD at diagnosis. Six patients (60%) were staged as T4 disease and three patients (30%) N1 nodal status. 

### 3.2. Neoadjuvant Treatment and Response

All patients were treated with neoadjuvant chemotherapy. Eight patients received folinic acid, fluorouracil, irinotecan and oxaliplatin (FOLFIRINOX) with 50% completing 12 cycles (Table 2). Dose adjustment or termination of treatment due to side effects was at the discretion of the patient’s lead consultant oncologist. One patient received gemcitabine and capecitabine due to poor tolerance to chemotherapy for a previous non-HPB malignancy, and one patient was treated with gemcitabine and oxaliplatin on a 2006 treatment protocol. Following neoadjuvant chemotherapy, the cross-sectional imaging and response were reviewed in the pancreatic MDT. Further neoadjuvant chemoradiation was recommended if the resection margin remained threatened or involved. Eight patients received further chemoradiation (Table 3). 

The mean pre-NAT ^18^FDG-PET/CT SUVmax and Ca19-9 were 6.14 ± 1.98 units and 148.0 ± 146.3 IU/L, respectively. NAT led to a significant mean reduction in Ca19-9 to 18.0 ± 18.7 IU/L, *p* = 0.01 (Figure 1). Post-NAT triple-phase CT was performed on all patients and was the main mode of imaging used to assess response to treatment. Post-NAT ^18^FDG-PET/CT was performed in three patients, two of which showed a complete PET/CT-metabolic response: an SUVmax of zero. The mean duration of NAT was 5.6 ± 1.7 months, and the mean time from completion of NAT to surgical resection was 13.1 ± 8.3 weeks.

### 3.3. Surgical Resection

Surgical resection was performed in patients who had shown favourable radiological and metabolic response following NAT. All cases were reviewed by our central pancreatic MDT. Surgery was also subject to the patient passing a consultant-led anaesthetic pre-assessment. Five patients underwent Whipple’s pancreaticoduodenectomy, three underwent Whipple’s pancreaticoduodenectomy with venous resection, and one patient underwent a two-staged operation coeliac axis artery resection with pancreatosplenectomy (Table 4). 

The mean postoperative critical care unit stay was 2.0 ± 0.8 days and the mean hospital length of stay was 11.8 ± 6.2 days. Upon histological analysis, all cases showed complete primary pathological response (ypT0) and all achieved clear margins (R0). The mean lymph node yield was 21.1 ± 10.4 nodes. Only one patient’s disease was reported as ypN1, with one lymph node involved. The remaining nine patients were ypN0. 

No patient received further adjuvant oncological treatment following MDT consensus. The patients were followed up every three months with a clinical consultation and blood tests, including tumour markers for the first two years, followed by six monthly follow-up thereafter. Cross-sectional imaging was performed on a six-monthly basis for the first two years and annually for the following three years. Additional investigations were performed at the lead clinician’s discretion.

### 3.4. Outcomes

At the time of analysis in November 2023, the current mean overall survival for the cohort was 26.6 ± 20.7 months. Within this case series, one patient developed pulmonary metastasis at 33 months and passed away at 35 months. The only other recurrence was reported at 16 months; this was isolated and within the surgical resection bed, and this patient did not undergo pre-operative chemoradiation. The patient’s recurrence was treated with chemoradiation, and they remain alive at the point of the most recent follow-up at 28 months. One patient had been lost to follow-up after discharge at 75 months. All other remaining patients are currently disease-free and remain under active follow-up at our institution. 

Two Clavien–Dindo grade III complications were reported in this series: one superficial wound dehiscence requiring VAC dressing, and a second patient requiring an intervention radiology-guided drain placed under local anaesthetic for an infected gallbladder fossa fluid collection. Specifically, there were no postoperative pancreatic fistulae. There were no Clavien–Dindo grade IV complications, no 30-day readmissions, and no 30-day mortalities reported in this series. 

## 4. Discussion

This retrospective case series describes a unique cohort of patients, which remains rare in the context of PDAC. Patients in this series were diagnosed with advanced disease at the time of presentation; not only did the patients reach a stage of resectability following neoadjuvant treatment, but they did so with no postoperative mortalities, a morbidity rate of Clavien–Dindo grade III or above of only 20%, and a length of stay post-surgery comparable to the published literature following major pancreatic resection [21]. All resections were R0, corresponding with complete response of the primary lesion following NAT. 

The primary endpoint of overall survival in this study was over two years at the point of analysis. There have been other series and case reports within the literature of pCR [16,22,23,24]. One larger national database study from the United States [17] included 5364 patients who underwent NAT for PDAC and reported a nationwide pCR rate of 0.8%. Other series reporting pCR following NAT for BD/LA-PDAC have all demonstrated an improved OS benefit [16,18,25], up to 76 months in one series [18]. As expected, those patients with a partial or no response following NAT did not benefit from any improvement in OS [26], which also implies aggressive tumour biology in these patients. 

Another one hundred patient series from the United States even reported a complete primary response rate of 14.5% in patients with BD-PDAC treated with neoadjuvant chemotherapy and stereotactic radiotherapy followed by surgery [27]. However, this particular series also reported N1 rates of 63.6%, implying that pCR does not always equate to disease control [27]. pCR, while associated with improved OS, does not equate to a cure. Our case series has reported a low recurrence rate so far, which compares to the reported rate of 33% within the literature following pCR [28]. Despite this, the reported 5-year survival rate was still approximately 70% [28], which is remarkable considering advanced disease at the time of diagnosis and is likely related to the close surveillance and aggressive treatment of disease recurrence in this cohort of patients, which confers further additional survival benefits [18]. A summary of these studies reporting pCR rates following NAT and surgery for PDAC are included below in Table 5.

Within this series, there was one case of disease recurrence at the surgical bed, which was found 16 months following surgery. This occurred in one of the two patients who did not receive neoadjuvant chemoradiation prior to surgery. The recurrence was treated with chemoradiation, and the patient has remained disease-free since. Our observation is that chemoradiation confers an additional oncological benefit, particularly in treating the primary site of disease, and is based on the theory that chemoradiation is targeted at the primary tumour, causing tumour necrosis and the sterilisation of the surgical field. However, whether this effect impacts the eventual OS is still widely debated and yet to be proven in a clinical trial setting. 

Several trials have aimed to investigate the role of radiotherapy and chemoradiotherapy in both resectable and locally advanced PDAC. The Lap07 trial reported no overall survival benefit but did report a 6-week progression-free benefit for patients treated with chemotherapy versus chemotherapy and definitive chemoradiation in locally advanced non-resectable PDAC [30]. The Conko-007 trial reported that neoadjuvant chemotherapy followed by chemoradiation in LA PDAC improved rates of complete resection margins (R0), but it was not evident from the results of the trial whether this had an impact on the eventual OS [31]. 

Chemoradiation has also been demonstrated in other non-clinical trial series to increase the likelihood of complete or near-complete pathological response, or improved R0 resection rates with an ensuing OS benefit [32,33]. Another propensity score-adjusted analysis from MD Anderson comparing patients with resectable PDAC who received neoadjuvant chemotherapy versus chemoradiation alone over a 15-year span found pre-operative chemoradiation to be associated with increased R0 rates and less lymph node positivity but found similar OS between the two groups [34]; this study did not report on any patients who underwent combination NAT. 

Trial results from the approach of using chemotherapy to treat PDAC as a systemic disease, followed by selective chemoradiation to the primary disease site in BD where the margin remains threatened post-chemotherapy, or in patients with LA disease to achieve maximal downstaging prior to surgery, are not clear cut yet. This series adds to the growing literature that chemotherapy followed by chemoradiation as a combined aggressive neoadjuvant approach may give selected patients with locally advanced disease the best chance of reaching resectability and a potential chance at a cure. 

This series also highlights that further research is required to identify both patient- and tumour-specific factors that define patients who are more likely to develop a pCR. For example, one small study reported BRCA-2 germline mutation carriers to have a higher chance of pCR following NAT, with an underlying theory that platinum-based chemotherapy has benefits in the early course of the disease [35]. Identifying similar molecular or genetic targets may lead to the development of other therapeutic agents. Younger patient age, lower baseline Ca19-9, as well as the use of gemcitabine as a radiosensitiser for chemoradiation have been associated with major pathological response following neoadjuvant treatment [36]. None of these parameters are highly sensitive or robust. Patients in the series demonstrated a significant Ca19-9 reduction following NAT, which has been shown to be a good marker for effective NAT and may even be a useful predictor of the incidence of liver metastases [37]. Measuring the Ca19-9 response should be considered as part of the assessment of response following NAT [38] for PDAC.

^18^FDG-PET/CT, in addition to a CT scan, is utilised routinely as part of pre-treatment staging investigations at our institution. The authors accept that this is not routine practice throughout the country and around the globe due to factors including the allocation of resources, the cost of such an investigation, as well the general availability of ^18^FDG-PET/CT. Only three patients in this series underwent restaging ^18^FDG-PET/CT in addition to a triple-phase CT scan following NAT, but it has been suggested by several groups that the utilisation of ^18^FDG-PET/CT in addition to CT is more useful in assessing the response and restaging of BD and LA PDAC following NAT when compared to CT alone [39,40]. 

Our series was based on postoperative histology analysis, but radiomics [41] and biomarkers such as circulating DNA-based technology may be utilised in the future to assess and monitor response to neoadjuvant treatment [42], for postoperative surveillance, and perhaps even predict those patients who may develop pCR allowing for a more refined, selective, and personalised oncological and surgical approach. 

There are several limitations to this study. This is a small, retrospective, single-institution descriptive case series without a comparison group. As the timeframe spanned over a decade, there have been adjustments to treatment pathways over time and, thus, the neoadjuvant treatment patients received was not identical. Patients also underwent surgery under the care of four different consultant surgeons over this timespan. The follow-up time frame has not reached five years for the majority of patients yet. Finally, the total number of patients who underwent NAT for PDAC at our institution over the study period is not available to enable the surgical resection rate and rate of pCR to be calculated and compared to the published literature. 

## 5. Conclusions

Primary pCR is uncommon but possible following NAT and leads to improved overall survival in patients with PDAC. Further work to identify the common denominator within this unique cohort may lead to advances in the personalised therapeutic approach for patients diagnosed with BD or LA PDAC. Achieving primary pCR has long-term overall survival benefits but does not equate to a cure. 

## Figures and Tables

**Figure 1 cancers-16-00452-f001:**
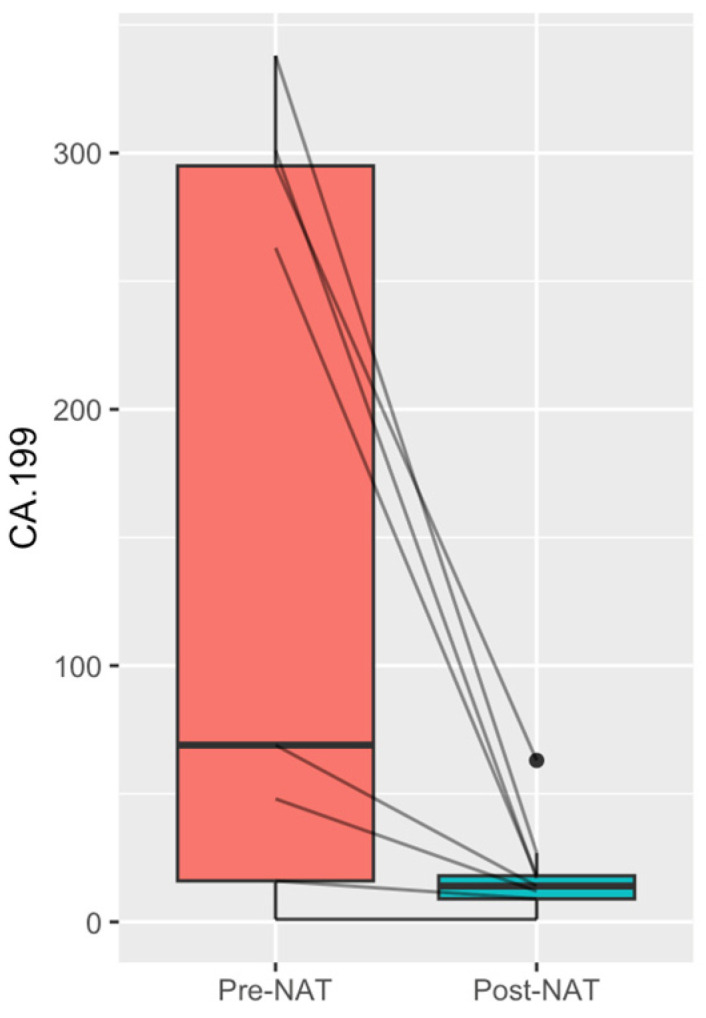
CA19-9 (IU/L): pre- and post-neoadjuvant treatment, *p* = 0.001.

**Table 1 cancers-16-00452-t001:** Case Series Patient Characteristics.

Patient Demographics	*n*	%
**Gender**		
Female	4	40
Male	6	60
**Age** (years)		
≤65	6	60
>65	4	40
**BMI** (mean ± sd, kg/m^2^)	22.89 ± 3.48	-
**NCCN Classification**		
Borderline	5	50
Locally Advanced	5	50
**T staging**		
T2	3	30
T3	1	10
T4	6	60
**N Staging**		
N0	7	70
N1	3	30

**Table 2 cancers-16-00452-t002:** Causes of early termination of FOLFIRINOX.

Patient	Cycles of FOLFIRINOX	Cause
1	11	Thrombocytopenia
2	7	Hair loss, skin rash
3	4	Typhlitis and neutropenia requiring GCSF
4	8	Neutropenia requiring GCSF

GCSF = granulocyte colony-stimulating factor.

**Table 3 cancers-16-00452-t003:** Chemoradiation regimens.

Chemoradiation Regimen	*n*	%
**2006**		
45Gy/25# with Capecitabine	1	10
**2011**		
54Gy/30# with Capecitabine	1	10
**2019 onwards**		
45Gy/15# with Capecitabine	6	60

**Table 4 cancers-16-00452-t004:** Surgical resection procedures.

Operation	*n*	%
Whipple’s	3	30
Robotic Whipple’s	2	20
Whipple’s with Venous Resection	3	30
Distal Pancreatectomy with Splenectomy	1	10
SCARPS	1	10

SCARPS = staged coeliac artery resection with pancreatosplenectomy [20].

**Table 5 cancers-16-00452-t005:** Summary of reported complete pathological response rates and overall survival following neoadjuvant treatment and surgery.

Authors	Year	*n*	pCR Rate	Reported Mean or Median OS
Blair et al. [18]	2021	331	9%	76 months
Sell et al. [17]	2020	5364	0.8%	43 months
He et al. [16]	2018	186	10%	>60 months
Rashid et al. [27]	2016	121	14.5%	pCR Specific OS N/R
Rose et al. [29]	2014	64	10%	pCR Specific OS N/R

*n* = number of patients included in study; pCR = complete pathological response; OS = survival; N/R = not reported.

## Data Availability

Data presented are contained within the article. The authors can be contacted for data; requests will only be considered for academic collaboration purposes.

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
