# Peer review of "Complete Primary Pathological Response Following Neoadjuvant Treatment and Radical Resection for Pancreatic Ductal Adenocarcinoma"

_cancers, 2024, doi:10.3390/cancers16020452_

Round 1
Reviewer 1 Report
Comments and Suggestions for Authors I read with great interest this single-centre case serie on locally advanced/borderline PDAC, undergone neoadjuvant treatment and subsequently surgery, in over almost two decades (from 2006 to 2023). The paper clearly report the primary and secondary endopoints of this retrospective review of a prospectively maintained database: OS and surgical outcomes respectively. The results are clearly described. In particular all the patients achieved a complete pathological response and a R0 surgery; complications are reported and overall survival had a quite wide range 26.6 +/- 20.7 months. This study reports a particular subtype of patients affected by advanced PDAC who clearly benefits from NAT (either chemo or chemo-radiotherapy). The advantages of neoadjuvant chemotherapies could be related to a secondary patient selection according to tumour biology. Traditionally, chemoradiation schemes had been administered in BR/LA-PDAC, but the resection rates did not reach 30%. In recent years, the introduction of FOLFIRINOX scheme and its modification as neoadjuvant therapy for BR-PDAC, has demonstrated to be highly effective, with a resection rate variably improved to 50%-80%. In this regard, ss this study includes only ten patients, a single table with patients baseline characteristics-type of NAT-surgery-complications-alive/death-cause of death would have been more useful. The limitations are clearly stated in the discussion. There are no images to reviewAuthor Response
We thank reviewer 1 for the summary and comments.
We would be inclined to reason that a further table in the suggested format would not be useful. Patient’s baseline characteristics are listed in Table 1, the neoadjuvant treatment regimens are described in detail in section 3.2 and finally at the time of analysis there was only one death (secton 3.4) a further table stating this would not provide additional benefit.
Reviewer 2 Report
Comments and Suggestions for Authors
The title ABSTRACT is missing
In the abstract, many abbreviations should be fully expressed before using abbreviations, such as SUVmax, CT, 18FDG-PET/CT
Says: only chance a cure
Should be: only chance for a cure
It would be interesting to add the total number of surgical PDAC cases from which these 10 patients were selected. Other important issue is to know how many patients with border line or locally advanced disease responded to neoadjuvant chemoradiation.
It would also be interesting to know if all the patients were operated by the same team.
Author Response
We thank reviewer 2 for the comments.
- Suggested abbreviation corrections and minor language changes now included.
-
We agree it would be very interesting to know the total number of sugical PDAC cases and PDAC cases who have undergone neoadjuvant treatment and not proceeded to surgery. Unfortunately as acknowledged in the limitations paragraph, this was not avalable. Minor revision in the limitations chapter to highlight this point.
-
Finally as the time spanned over 17 years, the patients were under care of four different consultant surgeons. This information has also been added to the limitations paragrah.
Reviewer 3 Report
Comments and Suggestions for Authors
Congratulations for article.
I want at discussion a table with comparation with other studies from literature.
Author Response
We thank reviewer 3 for the comments, a table has been added in the discussion section (table 5).